# Atomic-Scale Structural Properties in NiCo_2_O_4_/CuFe_2_O_4_ Bilayer Heterostructures on (001)-MgAl_2_O_4_ Substrate Regulated by Film Thickness

**DOI:** 10.3390/ma17040871

**Published:** 2024-02-13

**Authors:** Kun Liu, Ruyi Zhang, Jiankang Li, Songyou Zhang

**Affiliations:** 1School of Electronics and Information Engineering, Suzhou Vocational University, Suzhou 215104, China; ljk@jssvc.edu.cn (J.L.); 92207@jssvc.edu.cn (S.Z.); 2Ningbo Institute of Materials Technology and Engineering, Chinese Academy of Sciences, Ningbo 315201, China; zhangruyi@nimte.ac.cn

**Keywords:** bilayer heterostructure, film thickness, microstructural defects, electron microscopy

## Abstract

Changing film thickness to manipulate microstructural properties has been considered as a potential method in practical application. Here, we report that atomic-scale structural properties are regulated by film thickness in an NiC_O2_O_4_(NCO)/CuFe_2_O_4_(CFO) bilayer heterostructure prepared on (001)-MgAl_2_O_4_ (MAO) substrate by means of aberration-corrected scanning transmission electron microscopy (STEM). The misfit dislocations at the NCO/CFO interface and antiphase boundaries (APBs) bound to dislocations within the films are both found in NCO (40 nm)/CFO (40 nm)/MAO heterostructures, contributing to the relaxation of mismatch lattice strain. In addition, the non-overlapping *a*/4[101]-APB is found and the structural transformation of this kind of APB is resolved at the atomic scale. In contrast, only the interfacial dislocations form at the interface without the formation of APBs within the films in NCO (10 nm)/CFO (40 nm)/MAO heterostructures. Our results provide evidence that the formation of microstructural defects can be regulated by changing film thickness to tune the magnetic properties of epitaxial bilayer spinel oxide films.

## 1. Introduction

The spinel AB_2_O_4_ is widely used in microwave devices, spin filter devices and gas sensors due to its superior properties of high saturation magnetization, high Curie temperature, high spin polarizability, high permeability, high resistivity and small loss at high frequencies [1,2,3,4]. CuFe_2_O_4_ (CFO) has drawn a lot of interest among spinel ferrites because of its distinct physical and structural characteristics. In the CFO spinel structure, Cu^2+^ ions prefer to occupy octahedral (B-sites) sites, while the Fe^3+^ ions occupy both tetrahedral (A-sites) and the rest of the octahedral (B-sites) sites. The bulk CFO with the cubic phase (space group Fd$\bar{3}$m) has a lattice parameter of 0.840 nm. It has been discovered that the deposition conditions of CFO thin films have an impact on the cation distribution and phase stability of CFO films, which in turn has a significant effect on the films’ magnetic, electrical and optical properties [5,6]. It has been reported that CFO thin films deposited by pulsed laser deposition present the antiferromagnetic α-Fe_2_O_4_ phase along with the single-cubic spinel phase, and the increased ratio of hematite to magnetite phases altered by elevated annealing substrate temperature results in the saturation magnetization value decreasing while the coercivity value increases [7]. Another thin film layer of the bilayer heterojunction system is NiC_O2_O_4_ (NCO), which is an inverse spinel structure, i.e., half of the Co^3+^ ions occupy tetrahedra and the remaining Co^3+^ and Ni^2+^ ions occupy octahedra, with a cubic phase structure (space group Fd$\bar{3}$m) and lattice constant of 0.811 nm. It is well known that both electrical and magnetic properties are closely related to the film growth conditions in film systems. Substrate temperature, film thickness and oxygen pressure are important factors to tune the valence of anions and the occupancy distribution of anionsin the polyhedra, further manipulating the strength of exchange interactions and ferromagnetic order, which in turn affects the general physical properties in NCO film systems [8,9,10,11,12]. The NiCo_2_O_4_ (111)/Al_2_O_3_ (001) film system possesses semiconductivity and great magnetoresistance, which cannot be accounted for by spinel inversion and valence mixing, but by the appearance of antiphase boundaries (APBs) between nanosized crystallites, resulting from the structural mismatch between NiCo_2_O_4_ and Al_2_O_3_ [13].

The NCO and CFO films grown on MgAl_2_O_4_ (MAO) (001) substrate formed a bilayer subferromagnetic heterojunction, further generating an exchange bias effect, which plays a crucial role in spintronics devices [14], also occurring in the spinel oxide system [15,16,17]. The factors affecting the exchange bias effect are the respective structural characteristics and magnetic properties of the bilayer films, as well as the microstructure of the interface [18,19,20]. Particularly, the antiferromagnetic APBs normally form in the spinel film, in which the density and distribution of APBs would effectively affect the general magnetic properties [21,22]. It is imperative to manipulate the production of APBs in a reliable and controllable manner for innovative device applications [23,24,25]. Previous studies on the exchange bias effect have mostly focused on alloy and perovskite heterojunctions [26,27,28,29,30]. However, the microstructural properties related to the exchange bias effect in bilayer spinel oxide heterojunctions have rarely been reported. Here, we focus on the atomic-scale structural properties of spinel bilayer heterostructure regulated by film thickness.

In the present work, the atomic-scale structural properties at interfaces and within the films in NCO/CFO bilayer heterostructures with different top film thickness were investigated by means of aberration-corrected scanning transmission electron microscopy. The structural transform of APBs was resolved by the high-resolution high angle annular dark field (HAADF) imaging technique. The behaviors of strain relaxation between bilayer films were characterized. Particularly, the results in the present work illustrate that atomic-scale structural properties as well as the formation of APBs can be regulated by film thickness in spinel bilayer heterostructures.

## 2. Materials and Methods

Firstly, the CFO thin films were fabricated on single-crystal MAO (001) substrates by radio-frequency magnetron sputtering deposition. After that, the NCO thin film was deposited on the CFO thin film by radio-frequency magnetron sputtering, forming the bilayer heterostructures. The MAO (001) substrates, polished on one side with a surface roughness lower than 0.5 nm, were purchased from MTI Corporation (Richmond, Canada) and had a cubic phase structure (space group Fd$\bar{3}$m) with a lattice parameter of 0.808 nm. The distance between target and substrate was 15 cm and the base pressure of the deposition chamber was set as 10^−5^ Pa. The substrate temperatures during the deposition of the bottom CFO films and top NCO films were both 400 °C. The growth of CFO films and NCO films was carried out under the pressure of 50 Pa and 75 Pa, respectively, with an atmosphere mixture of O_2_ (99.99%) and Ar_2_ (99.99%) at the ratio of 1:1. The power density of 5.1 W/cm^2^ was applied on the target and the sputtering speed of CFO and NCO thin films was 2.74 nm/h and 3.84 nm/h, respectively. Two samples with different top film thickness were investigated in this paper. One sample was NCO (40 nm)/CFO (40 nm)/MAO (001), and the other sample was NCO (10 nm)/CFO (40 nm)/MAO (001).

The preparation of two (S)TEM specimens was completed by the focused ion beam (FIB) lift-out technique on an FEI Helios600i FIB/SEM instrument (Thermo Fisher Scientific, Waltham, MA, USA). FIB lamellae were cut along the <110> crystal orientations of the MAO substrate. TEM images were performed on the FEI Titan 80-300 microscope operated at 300 kV. STEM-HAADF imaging was carried out on a JEOL-ARM200F equipped ((JEOL Ltd., Tokyo, Japan) with a probe aberration corrector, operated at 200 kV. In STEM-HAADF imaging mode, a probe size of 0.1 nm with a semi-convergence angle of α = 22 mrad was used. The HAADF detectors covered angular ranges of 90~176 mrad.

## 3. Results and Discussions

The low-magnification bright field TEM (BF-TEM) image in Figure 1a shows the cross-sectional overview of the NCO (40 nm)/CFO (40 nm) film grown on MAO (001) substrate, recorded along the [1$\bar{1}$0] zone axis of the MAO. The thicknesses of the NCO/CFO film were measured to both be about 40 nm from Figure 1a. The film interfaces were distinguished to be visible according to contrast variation, as indicated by two horizontal white arrows. Figure 1b shows a standard selected area electron diffraction (SAED) pattern of the NCO/CFO film and part of the MAO substrate, recorded along the [1$\bar{1}$0] zone axis of the MAO substrate. The SAED pattern suggests that bilayer films on MAO (001) substrate both present cubic phases. The splitting of the 026 and 004 reflections along out-of-plane directions was discerned, indicating that the mismatch strain between film and substrate relaxed along out-of-plane. In contrast, the splitting of the 440 diffraction reflection along in-plane was not observed, as shown by the magnified part in the insert in Figure 1b. Particularly, the reflection spot splitting between bilayer films was neither observed along in-plane or out-of-plane. Figure 1c–e are energy dispersive spectroscopy (EDS) mapping images acquired from the regions marked by the white square in Figure 1a, representing the signals of Co–Kα1, Fe–Kα1 and Al–Kα1, respectively. The EDS results imply that the NCO film is fabricated on the top of the CFO film. From the result of SAED, it seems that the relaxation of the epitaxial mismatch strain between two films along in-plane does not occur. Further investigation needs to be performed to distinguish the microstructures within the bilayer films and across the interfaces.

The bilayer heterostructure of NCO (40 nm)/CFO (40 nm)/MAO is further examined by atomic resolution HAADF imaging, as displayed in Figure 2, recorded along the [1$\bar{1}$0] zone axis of the MAO. Based on the principle of Z contrast in HAADF imaging and element atom number (Z_Ni_ = 28, Z_Co_ = 27, Z_Cu_ = 29, Z_Fe_ = 26), the brighter spots in the HAADF image are caused by the higher atom column density, which is two times that of the darker spots along the [1$\bar{1}$0] zone axis [31]. The HAADF images in Figure 2a,b contain the interface between CFO film and MAO substrate, and NCO film and CFO film, respectively, which shows the epitaxial interfaces at parts of regions and that no interfacial dislocations form at this region.

As a contrast, the high resolution HAADF images were acquired from other regions across the interface between the NCO film and CFO film, recorded along the [1$\bar{1}$0] zone axis of the MAO substrate, as shown in Figure 3. The interface between the NCO film and CFO film was denoted by the horizontal white dashed line. Remarkably, the interfacial dislocations and planar defects were both observed. The image in Figure 3a contains an interfacial dislocation with the projected displacement of *a*_n_/8<112> (*a*_n_ is a lattice parameter of NCO film), acquired by the Burger circuit. This dislocation then induced an APB propagating into the NCO film, as indicated by the oblique white arrow. The APB was located at the (11$\bar{1}$) plane and composed of octahedral B-site cations of the two domain regions separated by the APB. Neither overlapping of the two domains nor a blurred area across the APB were observed. For clarity, this kind of APB is called a non-overlapping APB-I.

Figure 3b suggests that an *a*_c_/4[110]-APB, which is edge-on along the [1$\bar{1}$0] viewing direction, marked by the white vertical arrow, formed within the CFO film near the interface. The APB penetrated across the interface between the NCO film and CFO film, propagated into the NCO film and decomposed into two APBs with displacement vectors of *a*_n_/4[01$\bar{1}$] and *a*_n_/4[101], respectively, marked by two oblique white arrows. These two APBs had the projected displacement vector *a*_n_/8<112> along the [1$\bar{1}$0] direction. The regions across the two APBs were blurred, both of which were the overlapping regions of the two domains, which was also observed in our previous work [32]. In addition, a dislocation with the projected displacement vector of *a*_n_/4<00$\bar{1}$> existed at the interface acquired by the Burger circuit. The dislocation core lay within the NCO film, which was connected with the APB of *a*_n_/4[01$\bar{1}$] in the NCO film. Eventually, the dislocation terminated at this APB.

Figure 3c shows that an edge-on APB with the displacement of *a*_c_/4[110] formed within the CFO film near the interface, as marked by the lower white arrow, which crossed the interface, penetrated into the NCO film and translated along the [$\bar{1}\bar{1}$0] crystal direction, as marked by the upper white arrow. It is noticeable that there was no formation of interfacial dislocation in this region by the closed Burger circuit containing the interface and two APBs.

In the NCO (40 nm)/CFO (40 nm) heterostructure, interfacial dislocations and APBs within the film formed to relax the epitaxial mismatch strain between the NCO and CFO films. The heterostructure interface can both terminate the APBs formed in the CFO film and become the starting point of the APB formation in the NCO film.

In addition, the structural transformation of non-overlapping APBs was observed and resolved for the first time in the NCO films, which are shown in the HAADF images of Figure 4, recorded along the [1$\bar{1}$0] zone axis of the film. In Figure 4a, the white oblique arrow denotes the non-overlapping APB-I with the projected displacement of *a*_n_/8[112], which separated the film into domain I and domain II. It can be seen that the APB lies in the (11$\bar{1}$) plane and consists of B-site cations of both domains I and II. It should be noted that the present APB does not appear in the overlapped regions but presents a sharp boundary. In fact, the full displacement vector of this type of APB can be deduced to be *a*_n_/4[101] = *a*_n_/8[112] + *a*_n_/8[1$\bar{1}$0] (or *a*_n_/4[011] = *a*_n_/8[112] + *a*_n_/8[$\bar{1}$10]), taking account of the displacement along the [1$\bar{1}$0] zone axis, which suggests that the domains separated by this kind of APB are also displaced by 1/8 atomic plane in the [1$\bar{1}$0] viewing direction, eventually forming the non-overlapping APB-I. Particularly, this type of APB was also observed in the CFO film.

Figure 4b shows the atomic resolution HAADF image containing the translation of edge-on APBs. The two edge-on APBs both had the displacement vector of *a*_n_/4[110], recorded along the [1$\bar{1}$0] zone axis. The *a*_n_/4[110]-APB, denoted by the lower vertical arrow was displaced along the [112] crystal orientation and formed the other *a*_n_/4[110]-APB, denoted by the upper vertical arrow. The area circled by the oval dotted box was the connection point at the two APBs before and after the translation, as shown by the two connected brown diamonds in Figure 4b (the four vertices of the brown diamonds are the locations of the octahedral B-site cations). Remarkably, Figure 4c shows the continuous translation of *a*_n_/4[110]-APB along the [112] crystal orientation, denoted by a series of brown diamonds, which generated the domain boundary. This is called the non-overlapping APB-II for clarity, marked by an oblique white arrow. The non-overlapping APB-II also lies in the (11$\bar{1}$) plane and separates the film into domain I and domain II, and the boundary consists of the octahedral B-site cations of both domains I and II. However, it is different from the non-overlapping APB-I in Figure 4a. It appears that the non-overlapping APB-I in Figure 4a shifted by 1/8 atomic plane along the [112] crystal orientation would form the non-overlapping APB-II in Figure 4c. For a clearer explanation, further investigation can be performed using the structural models established by Vesta (version 3, Ibaraki, Japan) [33].

On the basis of the results in Figure 4 and the shift vector of *a*/4[101] = *a*/8[112] + *a*/8[1$\bar{1}$0], the atomic details of the translation and transformation of non-overlapping APBs can be understood by establishing the structural models along [1$\bar{1}$0], [010] and [101] crystal orientation, as displayed in Figure 5. In the structural models of APBs, the tetrahedral sites (A-sites) are featured in orange, and octahedral sites (B-sites) are featured in blue. Figure 5a displays the structural model of non-overlapping APB-I with displacement of *a*/4[101] projected along the [1$\bar{1}$0] viewing direction. It can be seen that the oxygen sublattice across the non-overlapping APB-I is sustained the same as that in the perfect AB_2_O_4_ spinel structure, but the periodic cation arrangement is disturbed. The non-overlapping APB-I is denoted by a red dashed line and lies in the (11$\bar{1}$) plane, which consists of only B-site cations of two separated domains, I and II. Then, rotating and seeing the structural model in the [010] viewing direction, as shown in the structural model of Figure 5b, the APB is marked by the red dotted line, while the displacement caused by this APB does not produce an observable difference at the APB in this viewing direction; as a result, the interruption of the cation sublattice at the APB could not be recognized. Continuously rotating the structural mode of Figure 5a and viewing it in the [10$\bar{1}$] direction, as shown in Figure 5c, it is obvious that the atomic structural model in Figure 5c is the same as the atomic arrangement in Figure 4c. The formation of the structural model in Figure 5c can be understood by the continuous translation of the edge-on *a*/4[101]-APB along the [121] crystal orientation, as shown in the brown diamonds and the connection point at the two APBs before and after the translation, which was also circled by the oval dotted box, and which is consistent with the HAADF result in Figure 4b. The edge-on *a*/4[101]-APB continuously translating along the [121] crystal orientation resulted in the formation of the non-overlapping APB-II. Consequently, the non-overlapping APB-I with the displacement of *a*/4[101] in the [1$\bar{1}$0] viewing orientation can be formed by the edge-on *a*/4[101]-APB in the [10$\bar{1}$] viewing orientation, continuously translating along the [121] crystal orientation, eventually making non-overlapping APB-I lie in the (11$\bar{1}$) habit plane. Obviously, the non-overlapping APB-I in Figure 5a projected in [1$\bar{1}$0] shifted by 1/8 atomic plane along the [112] crystal orientation would form the non-overlapping APB-II in Figure 5c projected in [10$\bar{1}$], which is consistent with the results of the HAADF experiment in Figure 4.

To additionally resolve atomic arrangement across the non-overlapping APB detected in the bilayers, the polyhedral connection model across this kind of APB is displayed in Figure 5d. In contrast to edge-sharing octahedra in perfect spinel lattices, across the non-overlapping APB, the edge-sharing as well as the corner-sharing octahedra coexist. For clarity, the octahedra at the APB are marked as B1, B2 and B3; in fact, the cations located at these octahedra have the same coordination environment as an oxygen octahedron. It can be seen that octahedra B1 and B3 (B2 and B3) share an edge with a bond angle of 90 degrees, and octahedra B1 and B2 share a corner with a bond angle of 180 degrees. As a result, a strong antiferromagnetic super-exchange interaction occurs across the non-overlapping APB [22], which is speculated to be an antiferromagnetic interface.

In order to understand the formation of dislocations and planar defects in different film thicknesses of NCO/CFO/MAO bilayer heterostructures, the NCO (10 nm)/CFO (40 nm) bilayer films were fabricated on the MAO (001) substrate under the same growth conditions. If the thickness of the NCO film changed, the heterojunction interface between the two films and the microstructures inside the film changed.

The cross-sectional overview of the NCO (10 nm)/CFO (40 nm) film fabricated on MAO (001) substrate, viewed along the [1$\bar{1}$0] zone axis of the MAO, is shown in Figure 6a. The film interface between the NCO film and CFO film was also visible due to the contrast difference, as indicated by two white horizontal arrows. Figure 6b displays the standard SAED pattern of NCO/CFO film and partial MAO substrate, viewed along the [1$\bar{1}$0] zone axis of the MAO substrate. The spot splitting of the 026 and 004 reflections along out-of-plane directions was discerned, while the splitting of the 440 diffraction reflection along the in-plane was not observed. Particularly, the reflection splitting of bilayer films was observed neither along the in-plane nor the out-of-plane. High resolution HAADF imaging was further conducted at the interface of the bilayer film. Figure 6c displays the smooth epitaxial interfaces, marked by a horizontal white line, without the formation of interfacial dislocations in this region. In contrast, the interfacial misfit dislocations with different Burger vectors were found at the interfaces in other interfacial regions.

Figure 7a displays an atomic-resolution HAADF image of the NCO (10 nm)/CFO (40 nm) interface, recorded along the [1$\bar{1}$0] zone axis of the MAO substrate, displaying the existence of interfacial dislocations. Conducting a Burger circuit around the misfit dislocation core produced the projected displacement vector of *a*_c_/2<001>. It is obvious that the dislocation core, marked by the white arrow, is located inside the NCO film near the interface. In addition, another kind of interfacial dislocation with the projected displacement vector of *a*_c_/2<001> + *a*_c_/8<112> was observed at the interface shown in Figure 7b, acquired by the Burger circuit containing two dislocation cores, one core located inside the NCO film adjacent to the interface and another core located at the interface, as indicated by two white arrows in Figure 7b. It should be noted that neither of these two dislocations led to the formation of APBs in two films, which is different from NCO (40 nm)/CFO (40 nm)/MAO heterostructures. The formation of interfacial dislocation contributes to the relaxation of epitaxial mismatch strain between NCO and CFO films.

Two NCO/CFO/MAO heterosystems with different film thickness were prepared under the same deposition conditions. Different microstructural defects were generated inside and at the interface of the two films due to the different thickness of the upper film NCO.

In NCO/CFO bilayer heterostructures, the in-plane lattice mismatch between NCO and CFO materials was calculated as −3.45% by using (*a*_N_–*a*_C_)/*a*_C_*100%, where *a*_N_ and *a*_C_ are lattice parameters of the bulk NCO and CFO materials, respectively. It is noted that the strain state in the film-substrate heterostructure can be influenced by the film’s thickness [34]. The formation of both interfacial dislocations and APBs within the film in the NCO (40 nm)/CFO (40 nm) heterostructure contributed to epitaxial strain relaxation from lattice mismatch more effectively than that in the NCO (10 nm)/CFO (40 nm) heterostructure. The density of the interfacial dislocations and APBs changed with the thickness of film, which is also proved by the density of interfacial dislocation in the NCO (40 nm)/CFO (40 nm) heterosystem being greater than that in the NCO (10 nm)/CFO (40 nm) heterosystem. In some cases, the interfacial misfit dislocations could be the nucleation sites of the APBs, which indicates that the density of APBs relies on the density of interfacial misfit dislocations in the NCO (40 nm)/CFO (40 nm) heterostructure. Remarkably, the APBs including edge-on APBs and non-overlapping APBs form only in the NCO (40 nm)/CFO (40 nm) rather than the NCO (10 nm)/CFO (40 nm) heterostructure. Therefore, the formation of APBs can be controlled by film thickness in the present bilayer heterostructure. In fact, the controllable formation of APBs investigated by the high resolution HAADF imaging technique also be reported in spinel and perovskite heterostructures. The *a*/4[110]-APBs and overlapping APBs with displacement of *a*/4[011] bound to interfacial dislocations form in LiFe_5_O_8_ film, and the density of APBs can be controlled by the LiFe_5_O_8_ film thickness [23]. It has been proved that the formation of APBs originates from dislocations in the layered Li(Ni_1−x−y_Co_x_Mn_y_)O_2_ material [24]. The APBs naturally nucleate at the interface step by atomic control of heterointerfaces in the La_2/3_Sr_1/3_MnO_3_/Sr_2_RuO_4_ film system [35], which improves our understanding of controllable defect nucleation and formation to optimize physical properties in film.

In bilayer heterostructures, the microstructural properties, interface defects and further magnetic properties could be manipulated by film thickness [36,37,38,39]. Particularly, the exchange bias effect related to magnetic anisotropy and spin pinning occurring in bilayer heterostructures is dependent on the structural properties of magnetic film and interface roughness [27,40,41]. The interfacial dislocations, edge-on *a*/4[110]-APBs and non-overlapping *a*/4[101]-APBs form in the NCO (40nm)/CFO (40 nm) heterostructure. A single *a*/4[101]-APB with the form of a rock salt structure interlayer was reported in the NiFe_2_O_4_ film [22]. The APBs interrupt the perfect spinel lattice, and the cation configuration at the APB is different from that in the spinel structure. Abundant corner-sharing connections appear between B-site cations via oxygens with a bond angle of 180 degrees across the non-overlapping APBs, which leads to the APBs presenting antiferromagnetic coupling [21]. The formation of the APBs in NCO (40 nm)/CFO (40 nm) heterostructures brings about larger magnetoresistance, nonsaturation of magnetization and enhanced magnetic anisotropy [42,43,44,45] compared to that in NCO (10 nm)/CFO (40 nm) heterostructures. The formation of APBs may lead to the appearance of exchange bias at room temperature, and the strength of exchange bias field was found to increase with the increase in density of APBs [46,47]. The present result indicates that atomic-scale structural characterization of interfaces and APBs can be accomplished by advanced microscopy and the formation and density of APBs can be regulated by film thickness in bilayer heterostructures, which paves the way for the development of spinel oxide-based magnetic spintronic devices.

## 4. Conclusions

The microstructural properties of NCO/CFO/MAO bilayer heterostructures regulated by film thickness have been characterized at the atomic scale using aberration-corrected STEM. The interfacial dislocations of different Burger vectors form at the interface and antiferromagnetic APBs bound to the dislocations are generated inside the film in the NCO (40 nm)/CFO (40 nm) heterostructures. The non-overlapping APBs were found, and the structural transform of non-overlapping APBs was resolved at the atomic scale. In contrast, in the NCO (10 nm)/CFO (40 nm) heterosystems, only interfacial dislocations of different Burger vectors form at the NCO/CFO interface, which does not further cause the formation of APBs in films. The formation of interfacial dislocations and APBs in NCO (40 nm)/CFO (40 nm) heterostructures contribute to the more effective relaxation of mismatch strain. Therefore, our results demonstrate that changing the thickness of the top layer of the bilayer heterostructure can manipulate the microstructure at the interface and the formation of APB inside the film and further influence the physical properties (e.g., exchange bias effects and magnetoresistance).

## Figures and Tables

**Figure 1 materials-17-00871-f001:**
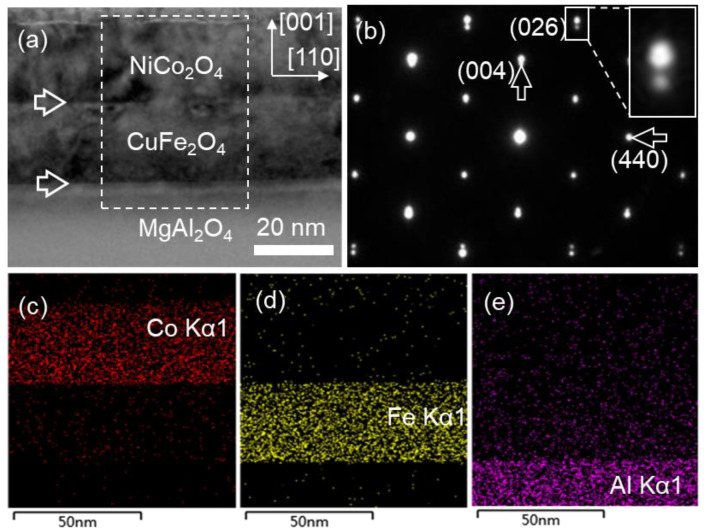
(**a**) Low-magnification BF-TEM images of NCO/CFO film on MAO (001) substrate. The two kinds of different interface are denoted by white horizontal arrows. (**b**) The SAED pattern of bilayer heterostructure film, recorded along the [1$\bar{1}$0] MAO zone axis. The splitting of the reflection spots is indicated by the vertical arrows. (**c**–**e**) EDS mapping of Co, Fe elements in film and Al element in substrate, respectively.

**Figure 2 materials-17-00871-f002:**
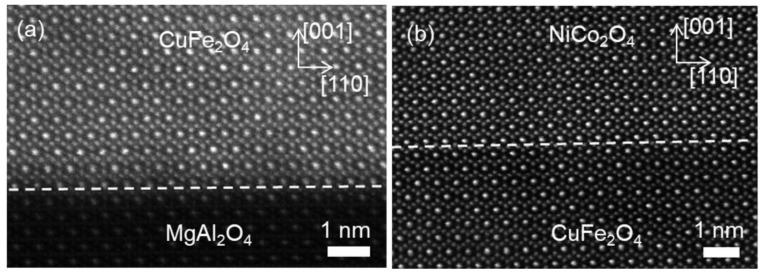
(**a**,**b**) Atomic-resolution STEM-HAADF images of the interface between CFO film and MAO substrate and the heterostructure interface of NCO film and CFO film, respectively, denoted by a white dashed line, show epitaxially coherent growth on substrate at this region.

**Figure 3 materials-17-00871-f003:**
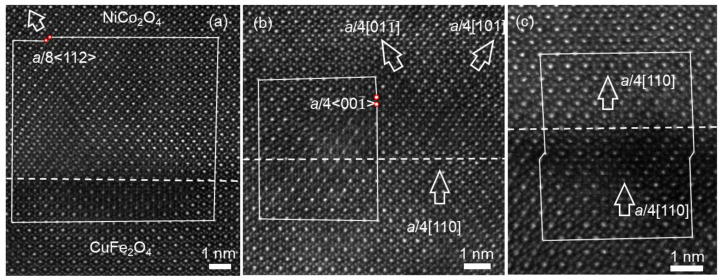
(**a**–**c**) Atomic-resolution STEM-HAADF images of the heterostructure interfaces between NCO film and CFO film in different regions, denoted by white dashed lines, showing the interfacial dislocations with different Burger vectors. The oblique arrows mark the APBs in the films. The red circles in (**a**,**b**) represent the start and end of the Burger circuit.

**Figure 4 materials-17-00871-f004:**
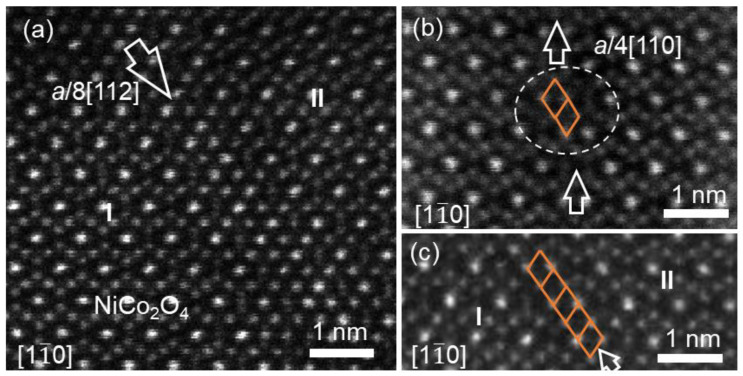
(**a**–**c**) Atomic-resolution STEM-HAADF images of NCO film in different regions, showing the non-overlapping APB located in (11$\bar{1}$) plane. The white arrows mark the APBs in the films. The orange lozenges represent the shift of the APB.

**Figure 5 materials-17-00871-f005:**
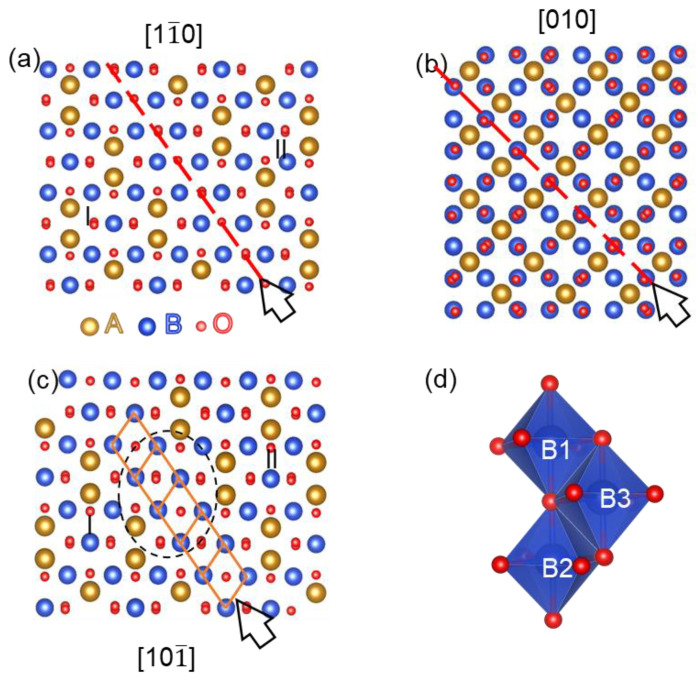
(**a**,**c**) The structural models of APB, observed along the [1$\bar{1}$0] MAO zone axis. The brown diamonds in (c) represent the continuous translation of edge-on *a*/4[101]-APB along the [121] crystal orientation. (b) The structural models of APB, observed along the [010] MAO zone axis. The non-overlapping APB plane is marked by a red dashed line. (d) The structural model showing the corner-shared octahedral cations at APB.

**Figure 6 materials-17-00871-f006:**
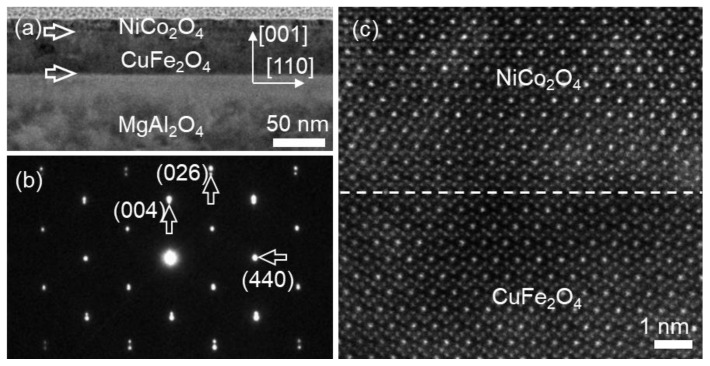
(**a**,**b**) Low-magnification BF-TEM images of NCO/CFO films on MAO (001) substrate. The two different kinds of interface are indicated by horizontal white arrows. (**b**) The typical SAED pattern of the bilayer heterostructure film, recorded along the [1$\bar{1}$0] MAO zone axes. The splitting of the reflection spots is indicated by the vertical arrows. (**c**) Atomic-resolution HAADF images of the interface between NCO film and CFO film denoted by a white dashed line, showing epitaxially coherent growth on substrate at this region.

**Figure 7 materials-17-00871-f007:**
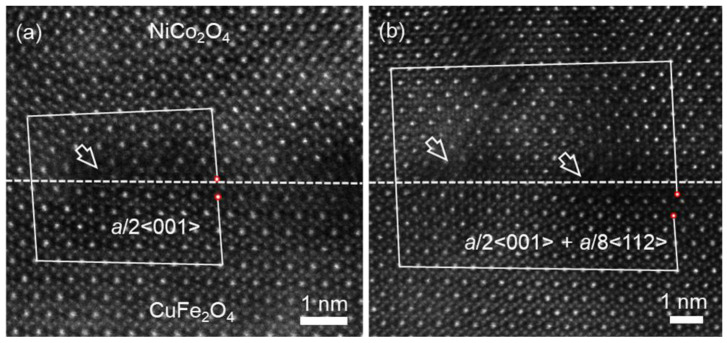
(**a**,**b**) Atomic-resolution STEM-HAADF images of the heterostructure interfaces between NCO film and CFO film in different regions denoted by white dashed lines, showing the interfacial dislocations with different Burgers vectors. The red circles represent the start and end of the Burger circuit.

## Data Availability

Data is contained within the article.

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
