# Peer review of "Atomic-Scale Structural Properties in NiCo2O4/CuFe2O4 Bilayer Heterostructures on (001)-MgAl2O4 Substrate Regulated by Film Thickness"

_materials, 2024, doi:10.3390/ma17040871_

Round 1

Reviewer 1 Report

Comments and Suggestions for Authors

The paper in which electron-microscopy studies of NiCo2O4/CuFe2O4 bilayer heterostructures sputtered on (001)-MgAl2O4 substrate are carried out at a very good level. The NiCo2O4 and CuFe2O4 structures are of interest primarily for the unique physical properties of these phases.

A global remark - this paper presents studies of structural properties only and there is no information about the studies of physical properties of these systems. Thus, the question arises - what is the purpose of these studies? The conclusion made by the authors that the influence of layer thickness on microstructure is observed is generally obvious and expected. Since the parameters of crystal lattices of NiCo2O4, CuFe2O4, MgAl2O4 are different, then in the process of epitaxial growth stresses will arise and structure defects will appear as a consequence.

The aim of the journal is: "the in-depth understanding of the relationship between structure, properties, and functions of all kinds of materials". Therefore, the physical properties are sorely lacking in the paper, which would show how defects arising from changes in the thickness of the layers affect the physical properties.

 Specific remarks

 When first mentioned in the text of the article, it is necessary to provide a transcription of abbreviations. The authors have done this in the abstract, but it is not enough.

 Information on the space group and lattice parameters of all three phases (NiCo2O4, CuFe2O4, MgAl2O4) should be given.

 In the Introduction it is desirable to describe in more detail (with specific examples) how structural properties affect the physical properties of thin film systems NiCo2O4 and CuFe2O4.

 In the "Materials and Methods" section, information on the type of magnetron sputtering as well as on the sputtering speed should be provided.

 Correct font formatting (Line 14 - "CO" instead of "Co"), upper and lower indices (Lines 73-74, 100).

When describing diffraction reflections (e.g., Line 94), phase information must be given.

Comments on the Quality of English Language

English does not require major adjustments.

Reviewer 2 Report

Comments and Suggestions for Authors

The authors investigated the atomic-scale structural properties of NiCO2O4(NCO)/CuFe2O4(CFO) bilayer heterostructure on (001)-MgAl2O4 (MAO) substrate by using aberration-corrected scanning transmission electron microscopy (STEM).

They demonstrated that the microstructure and magnetic properties of the bilayer spinel oxide heterostructure could be tuned by changing the film thickness of NCO. This work has some merit, however, it should be improved.

The discussion section lacks references. The authors should compare their results with the existing in the literature even for other spinel oxide systems or other types of heterostructures.   

Comments on the Quality of English Language

Minor editing of English language are required.

Reviewer 3 Report

Comments and Suggestions for Authors

The authors submitted a perfectly-written manuscript, with a proper research design. Results are clearly presented, the writing language is adequate, and the results support conclusions. 

1. The authors should address the possible application of found novelty, or at least discuss it somehow.
2. The X-ray reflectometry technique might be useful in the characterization of the interphase boundary of the thin film. Did the authors think about its application?
3. How the formation of antiphase boundaries might affect the properties of bilayer film?

Round 2

Reviewer 1 Report

Comments and Suggestions for Authors

Need to add a transcription of acronyms (CFO, NCO, MAO, APB) when first mentioned in the Introduction.

Lines 86-87. "The power density of 5.1 W/cm^2 .... sputtering speed is 2.74 nm/h"

This is not a typo, really 5.1 W/cm^2 and 2.74 nm/hour? So the sputtering of one sample lasted about 20-30 hours?

Line 93. "HAADF imaging."

It might be better to write "STEM HAADF imaging".

Line 103. "selected area electron pattern (SAED) pattern"

should be "selected area electron diffraction (SAED) pattern"

Lines 133, 169, 202, 288. "Atomic-resolution HAADF images"

It might be better to write "Atomic-resolution STEM HAADF images".

Comments on the Quality of English Language

No serious complaints about the level of English language
